# OpenReview forum: "Collapse of Irrelevant Representations (CIR) Ensures Robust and Non-Disruptive LLM Unlearning"
_ICLR.cc/2026/Conference — ICLR 2026 Conference Withdrawn Submission_

### Official Review · Reviewer_1ufv · 2025-10-24

**Soundness:** 2
**Presentation:** 1
**Contribution:** 2
**Rating:** 2
**Confidence:** 3

**Summary:**

This paper proposed a novel unlearning method called Collapse of Irrelevant Representations (CIR), which collapses subspaces containing common representations, thus making the unlearning more effective and robust.

**Strengths:**

1. Analysis is comprehensive. The authors did a systematic analysis on the problems with current unlearning strategies, especially the study on the disruption of superficially similar facts, offering a new perspective in understanding the pitfalls of unlearning.
2. The method is novel. The authors propose CIR approach, which collapses irrelevant representations, and demonstrate promising results in terms of the unlearning quality and robustness.

**Weaknesses:**

1. Writing is poor. Many concepts are vague and many figures are hard to understand. Examples are listed below:

   1. In Figure 2, I don't understand why in many cases, the WMDP accuracy during fine-tuning attack never reaches the WMDP accuracy at 0.1% Disruption threshold, and how these "WMDP accuracy at 0.1% disruption threshold" are computed. Also, I don't understand where we can infer "only unlearning that happened after the disruption threshold can be reverted, and unlearning that happened without disruption remains robust," based on Figure 2, since there are no indicators on which traces represent the revert process, and how we can know which one is reversible. Overall, this figure is messy and hard to read.
    2.  In Figure 3, I don't understand the meaning of the columns inside the activations and gradients plot. What is the meaning of "a slide of activations"? Is it a row/column of the activation? I just feel the dimension is incorrect, since for Llama3.1-8B, the hiddenlayer feature dimension should be 4096. I also don't understand the meaning of red and green in this figure.
    3. In Figure 4, I don't understand the meaning of "an update successfully unlearns a paraphrased fact". How do we define a successful unlearning? I also don't understand the meaning of the scale of the color. Similarly, the shape of the matrix seems erroneous since the hidden matrix shape of Llama3.1-8B MLP layer should be (4096, 14336) (mlp.gate, mlp.up) (14336, 4096)(mlp.down)
    4. In Figure 5 and Figure 1(b), I don't understand why the curve is not injective -- sometimes one wikitext loss corresponds to multiple WMDP accuracy in the unlearning plot.
2. Experiments are oversimplified. Only evaluate on the LLama3.1 model, which is a small, outdated model. What's more, loss on wikitext is not a good metric of utility. We have a bunch of benchmarks that evaluate the model's utility from different perspectives (MCQ and open-ended). The authors should at least choose some of them (say 3 benchmarks) to strengthen their argument.

**Questions:**

I have listed my questions in Weakness section.

---

### Official Review · Reviewer_MM7y · 2025-10-31

**Soundness:** 2
**Presentation:** 2
**Contribution:** 2
**Rating:** 4
**Confidence:** 2

**Summary:**

This work identifies a root cause leading to the failure of unlearning or safety training methods in targeting too general representations to be unlearned. Based on their analysis, the authors propose a new approach, Collapse of Irrelevant Representations (CIR), that collapses activation subspaces by eliminating principal components to allow for more targeted unlearning and prevent the unlearning of general knowledge. The CIR method is evaluated on Llama-3.1 on bio-/cyber-hazardous facts datasets against the Circuit Breakers baseline.

**Strengths:**

- S1. This work identifies and studies the important problem of unlearning/safety methods failing to achieve the promised target performance.
- S2. The authors conduct an analysis of what is causing unlearning methods to fail and investigate what conditions lead to such results.
- S3. The CIR method is grounded in the initial analysis and is effective, according to the authors’ evaluation.

**Weaknesses:**

- W1. The analysis is very empirical and appears to lack statistical significance or theoretical insight.
- W2. The computational cost and scalability of CIR are unclear. The authors report <3 GPU-seconds per fact on an 8B model, but there is no discussion on how this scales with model size or dataset size. Without such discussion, it's hard to judge if CIR would be practical for larger models or real-world use.
- W3. The experimental evaluation is limited to a single model (LLaMA-3.1, 8B) and one dataset of hazardous facts. This narrow scope raises questions about the generality of the findings. Evaluating on additional models or datasets (or providing justification why it wasn’t done) would make the results more convincing.

**Questions:**

- Q1. Section 3’s analysis could be more rigorous and explicitly structured. Can the authors summarize it by explicitly mentioning: (a) what the hypotheses involved are; (b) what the properties of the setting they are investigating are; and (c) what experimentation, metrics, and analysis they performed to validate their hypotheses?
- Q2. Can the authors provide a more theoretical overview that explains the intuition behind their analysis and method, with sufficient references to the relevant literature?
- Q3. Can the authors elaborate on why they chose to base their entire evaluation on a single dataset and a single model? Would it be possible to extend the evaluation further to make it more convincing?
- Q4. What are the computational requirements of CIR for larger models or more data? Can the authors comment on how the PCA and collapse steps scale with model size, and whether there are any optimization tricks to keep it efficient?

---

### Official Review · Reviewer_JtCL · 2025-11-01

**Soundness:** 3
**Presentation:** 3
**Contribution:** 3
**Rating:** 2
**Confidence:** 4

**Summary:**

The paper proposes Collapse of Irrelevant Representations (CIR), a selective framework for Large Language Model (LLM) unlearning that is robust to fine-tuning attacks.

The authors first identify critical weaknesses with existing unlearning methods, noting that they indiscriminately disrupt general representations shared between harmful and benign knowledge. This, they argue, creates superficial unlearning: while models appear to forget on target data, they remain highly vulnerable to relearning attacks that easily repair these shared representations.

To address this, the paper introduces CIR, which performs representation-level purification of unlearning updates to make them highly specific. The framework contains two main components:

(1) Activation–Gradient PCA, which identifies common subspaces in MLP layers that encode irrelevant or general-purpose representations;
(2) Subspace Collapse, which removes these components from activations and gradients before weight updates, ensuring that unlearning targets only fact-specific directions.

**Strengths:**

The paper makes a significant contribution by moving beyond the general goal of minimizing disruption in unlearning and instead empirically identifying the precise point at which unlearning becomes unstable. Through extensive experiments, the authors show that once general performance degradation exceeds roughly 0.1%, the unlearned knowledge can be easily recovered through fine-tuning. This analysis, based on 50 unlearning runs, provides clear evidence that only unlearning achieved within the non-disruptive regime is robust, while unlearning in the disruptive phase is largely reversible. This insight explains the failure of previous methods that tolerated higher disruption levels and establishes a concrete design criterion for future work: robust unlearning requires maintaining near-zero disruption.

**Weaknesses:**

**1. From the perspective of experimental datasets**

The paper only evaluates its method on a single dataset (WMDP) and a single model, which greatly limits the reliability and generalizability of its conclusions. The authors should include experiments on additional models and datasets to demonstrate the robustness and consistency of their approach.

**2. From the baseline comparison perspective:**

Claiming that Circuit Breakers represents the best-performing baseline is not entirely fair (L23–24: “greater reduction in post-attack accuracy than the best baseline (Circuit Breakers)”). At a minimum, the authors should compare CIR against more recent unlearning methods, such as the RMU method introduced in the WMDP paper itself. The two baselines reported in this work perform rather poorly, making it unclear whether the observed recovery under fine-tuning attacks results from under-unlearning rather than attack robustness. Consequently, the current comparisons are insufficient to convincingly establish CIR’s superiority against relearning attacks.

**Questions:**

This paper provides a solid analysis of the disruption and relearning problem. If the authors could include more experiments and stronger baseline comparisons to further validate their method, I would be glad to raise my rating.

---

### Official Review · Reviewer_Pq1a · 2025-11-01

**Soundness:** 1
**Presentation:** 1
**Contribution:** 1
**Rating:** 0
**Confidence:** 5

**Summary:**

The paper claims that unlearning is irreversible even under finetuning attacks till a certain threshold of retain degradation.
Then the paper introduces a method "Collapse of irrelavant representations", where the main idea is removal of directions in the activation and gradient space during updates, which relevant to non-forget information.\
The authors perform this by first averaging these quantities over some corpus (for example the unlearning corpus), then finding the principal components of this, which are subtracted from the original activations.

**Strengths:**

The paper seems to have interesting ideas about ablating out common representations between the unlearning information and the retain information.\
However, it is not ready for publication due to lack of clarity and experimental evidence.

**Weaknesses:**

Several points are unclear in the paper
- In Figure 1, the claim that finetuning attacks only restore WMDP accuracy till the disruption threshold seems inaccurate to me. This is because I see several points where it crosses the y=x line. Kindly explain or provide more evidence for this claim.
- A more direct evidence of this claim will be  performing finetuning attacks directly on those checkpoints where the disruption threshold is still below 0.1%. Is it possible to show that there is no relearning for these models even across multiple runs ?
- Disruption or disruption threshold is never defined. When it is said , 0.1% retain set disruption or 5% disruption of the retain loss, what does that mean ?
- Later on, in Figure 3, disruption is defined as : "cosine similarity between the model’s update on the ”Paris” fact and the other evaluated fact". Is this the cosine similarity of the gradients between two different inputs ?
- What is the loss that is being used when computing the gradient in Figure 3 ?
- Please specify which layer, which model and detailed experimental setting for Fig 3.
- In Section 3.4, the terms control updates and retaining updates are not defined. This makes it extremely hard to follow.
- The statement - "most representations are not specific to the fact we are trying to unlearn, but more general." seems to be formed from Figure 3.  This seems premature because this is obtained from a heatmap of just few activations. It is not surprising that activations of similar prompts i.e. "The capital of {country} is" are similar.
- Please include utility measures. Currently the only thing measured is WIkitext loss, which is not enough to capture the retaining of utility.
- In Figure 1 and 5, it seems Gradient Difference is unable to unlearn properly. This is surprising considering that it is an aggressive method Please clarify this discrepancy.


I believe the overall idea is valuable, specifically from a difference-in-means [1] lens.\
I strongly urge the authors to improve on the current version with the goal of improving clarity.


[1] Arditi, Andy, et al. "Refusal in language models is mediated by a single direction." Advances in Neural Information Processing Systems 37 (2024): 136037-136083.

**Questions:**

Please see Weaknesses Section.

---

### Note · Authors · 2025-11-13

**Comment:**

From the reviews it is clear to us that the paper will require major rework, so we decided to withdraw from ICLR.

We thank the reviewers for their time, and will work to expand the experiments and improve presentation, based on their suggestions.

**Withdrawal Confirmation:**

I have read and agree with the venue's withdrawal policy on behalf of myself and my co-authors.